# Social and Economic Determinants of Life Expectancy at Birth in Eastern Europe

**DOI:** 10.3390/healthcare12111148

**Published:** 2024-06-05

**Authors:** Viorel Țarcă, Elena Țarcă, Mihaela Moscalu

**Affiliations:** 1Department of Preventive Medicine and Interdisciplinarity, “Grigore T. Popa” University of Medicine and Pharmacy, Universității Street No. 18, 700115 Iassy, Romania; viorel.tarca@umfiasi.ro (V.Ț.); mihaela.moscalu@umfiasi.ro (M.M.); 2Department of Surgery II—Pediatric Surgery, “Grigore T. Popa” University of Medicine and Pharmacy, 700115 Iasi, Romania

**Keywords:** environmental factors, health expenditure, GDP per capita, life expectancy, panel data, I10, I15, I12, I18, C23, J11, O50, Q56

## Abstract

Life expectancy at birth is considered a parameter of the social development, health system, or economic development of a country. We aimed to investigate the effects of GDP per capita (as the economic factor), health care expenditure, the number of medical doctors (as social factors), and CO_2_ emissions (as the environmental factor) on life expectancy. We used panel data analysis for 13 Eastern European countries over the 2000–2020 period. After performing the analysis, we used a cross-country fixed-effects panel (GLS with SUR weights). According to our model, a one percent increase in health expenditure (as % of GDP) increases life expectancy at birth by 0.376 years, whereas each additional medical doctor per 10,000 inhabitants increases life expectancy at birth by 0.088 years on average. At the same time, each additional 10,000 USD per capita each year would increase life expectancy at birth by 1.8 years on average. If CO_2_ emissions increase by 1 metric ton per capita, life expectancy at birth would decrease by 0.24 years, suggesting that higher carbon emissions are capable of reducing longevity. Every European country has to make special efforts to increase the life expectancy of its inhabitants by applying economic and health policies focused on the well-being of the population.

## 1. Introduction

In the last decades, life expectancy in Europe has increased a lot, but this increase is still unequal between countries with different levels of socioeconomic development. Life expectancy at birth (LEB) is one of the most relevant parameters of the social development, health system or economic development of a country. Every European country makes special efforts to increase the life expectancy of its inhabitants by applying economic and health policies focused on the well-being of the population [1,2]. A *sine qua non* condition of the general well-being of the population is health; the World Health Organization [3] defines it as “a state of complete physical, mental, and social well-being and not simply the absence of disease or infirmity”. In highly developed countries, life expectancy exceeds 75 years, while in underdeveloped countries life expectancy barely reaches 50 years [4]. Of course, in Europe risk factors such as the absence of running water, poor hygiene, very difficult access to the health system due to the long distances from hospitals, and the very low income per capita now belong to the distant past.

One of the most important achievements of human society in the last quarter of a millennium is the constant increase in the average LEB, in all countries, regardless of the level of development [5]. However, there are still significant differences between the developed and the developing world. Life expectancy can have a major positive impact on the main aspects of socioeconomic life that take place in a certain country, including female fertility, investment in human capital, health and pension expenditures and, last but not least, economic growth itself [6].

If in the year 1900, the average LEB of a newborn was only 32 years. LEB has increased by more than two times until now, reaching a value of 71 years in 2021 and probably a value of 77.2 years in 2050, according to the United Nations—World Population Prospects 2022. Currently, there are huge disparities in terms of average LEB, varying between 50 and 55 years in the countries of Central Africa (Nigeria, Chad, Central African Republic, and South Sudan) to 80–85 years in the countries of Western Europe, Israel, Canada, Australia, New Zealand, South Korea and Japan. The situation was similar in the year 2022 in South-Eastern Europe, where, with the exception of Slovenia and Cyprus, all other countries have average LEB levels below 80 years, Bulgaria being in the last position, very slightly below the worldwide average level of 71.5 years [7].

Increasing the economic growth and development in a country, highlighted by GDP per inhabitant, leads to the prolongation of longevity, obviously leading to an increase in average life expectancy for its citizens [8,9]. Most of the studies that investigated economic growth showed a positive association between life expectancy, as an indicator of the health of a population, and the general dynamics of the economy [10,11]. Furthermore, gross fixed capital formation, population growth, and LEB can explain long-term economic growth [12].

Several studies in medicine [13,14] show that the quality level of the environment is an important influencing factor for the general state of health in a country: air and water pollution, greenhouse gas emissions, depletion of natural resources and soils deterioration, are all capable of decreasing life expectancy [1,15,16,17,18]. According to World Bank data, the level of CO_2_ per inhabitant in the EU has dropped dramatically in the last 21 years, from 7.8 metric tons per capita in 2000 to only 5.5 metric tons per capita in 2020. In the same period, LEB in the EU increased by more than 4 years (from 77.1 years in 2000 to 81.3 years in 2019), falling slightly to 80.4 years in 2021 due to the COVID-19 pandemic.

It is already well known that in association with behavioral and social factors, the increase in health expenditures plays an important role as a determining factor in improving the living conditions of the population, inherently leading to an increase in average life expectancy [19,20]. Analyzing the evolution of global health expenditures for the last 21 years expressed as a percentage of GDP, we observe a constant increase from a level of 8.62% in 2000 to 10.89% in 2020. The trend was similar for the average LEB, which increased by approximately 4 years, from 68 years in 2000 to 72 years in 2020. The phenomenon was also present in the case of Eastern European countries, where the average LEB increased from 72.7 years in 2000 to 76.8 years in 2020, and health expenditures increased from 7.1% to 9.6% (https://data.worldbank.org/, accessed on 21 December 2023). After studying 28 European Union countries for a period of 10 years (2001–2011), Bilas et al. revealed that attained education level and GDP per capita are responsible for 72.6–82.6% of the differences in LEB (depending on the year of observation) [21].

An important question for the healthcare system is: Does increasing the number of primary care physicians lead to better health outcomes and higher life expectancy? Since 1980, in two European countries (Germany and Poland), improvements in the quality of the medical care system have been related to gains in LEB [22]. Joint research from Harvard, Stanford, and Boston Universities highlighted the fact that, in the United States, mortality rates were lower in areas with more primary care physicians [23]. Another study, conducted in Iran, identified the number of doctors per 10,000 population, food availability, literacy rate, total fertility rate, and GDP per capita as the most important factors related to LEB [24].

Our study addresses the effects of economic factors, social factors, and environmental factors on LEB by surveying 13 Eastern European countries over a 21-year period (2000–2020) and collecting data from multiple sources to create a panel data set for analysis. Since LEB is a demographic characteristic with multifactorial determinism, we considered the analysis of multiple socioeconomic factors, as well as the CO_2_ emission, as an expression of the degree of industrialization of Eastern European countries during the last 21 years.

## 2. Materials and Methods

### 2.1. Sources of Information

We collected data for variables from the World Bank and the World Health Organization databases data during the period of 2000–2020 for 13 Eastern European countries. To perform the analysis, we used the statistical package EViews, version 13, developed by IHS Markit Ltd., London, UK.

LEB (years), as the dependent variable for our model, indicates the average number of years a newborn would live if the main patterns of mortality at the time of its birth remained the same throughout its life [25].

The first independent variable, current health expenditures per capita, expressed as a percentage of GDP, is an estimation prepared by the World Health Organization which refers to healthcare goods and services consumed by an average person during each year. The second independent predictor is the number of medical doctors per 10,000 inhabitants, a variable that, according to the World Health Organization definition, includes specialist medical practitioners, generalists, and medical doctors not further defined, in a specific national and/or subnational area. These two independent social factors represent a standard for health care system in every country, and according to specialized literature have a significant positive effect on LEB [26,27,28,29].

The third predictor, GDP per capita (PPP constant 2017 international $), based on purchasing power parity, represents gross domestic product converted to international dollars using purchasing power parity rates. Data are in constant 2017 international dollars. According to the World Bank, an international dollar has the same purchasing power over GDP as the US dollar has in the United States. Purchasing power parity converts different currencies into a common currency and equalizes purchasing power by accounting for differences in price levels between economies during the conversion. Therefore, comparisons based on PPP do not distinguish between the relative price levels of different goods in the economy and are less sensitive to the potential volatility of market exchange rates.

The fourth independent variable, CO_2_ emissions (tons per capita), results from fossil fuel combustion and cement production. This includes carbon dioxide produced by solid, liquid, and gaseous fuel consumption and gas flaring. Emissions data are sourced from Climate Watch Historical GHG Emissions (1990–2020). 2023. Washington, DC: World Resources Institute. Carbon dioxide (CO_2_) accounts for the largest proportion of greenhouse gases responsible for global warming and climate change. For Central and Eastern European countries, the average level of CO_2_ emissions per inhabitant in 2020 was 5.3 metric tons per capita, this value exceeding by approximately 23% the average level recorded worldwide.

### 2.2. Research Hypotheses

In order to highlight the health level of a population, in the last decades, different statistical models were tested, which were used as the dependent variables: LEB, healthy life expectancy, and general or infant mortality rate [30,31]. The independent variables used were those directly related to the healthcare system and its effects and lifestyle, such as cardiovascular diseases, metabolic disorders, respiratory diseases, diet and physical activity, working conditions, and environmental factors [19,20,28].

The specialized literature of the last half-century has pointed out the direct relationship between the volume of resources employed in the health system and the longevity of the population [28,32].

Regarding the association between socioeconomic development and LEB, most of the studies that investigated economic growth showed a positive and strong relationship between life expectancy as an indicator of the health of a population, and the general dynamics of the economy [10,11]. GDP per capita has a decisive role in increasing LEB, increasing the general level of development in a country, and leading to greater longevity [33]. Economic growth has a positive impact on people’s health. When the average income of a particular country is high, the government can spend more resources on improving the quality of life and, thus, the health of the population [34].

Environmental quality, measured as metric tons per capita of CO_2_ emissions, has a negative effect on life expectancy, adversely affecting both the physical health and mental health of a population [35].

In our paper, the following research hypotheses were considered:

**Hypothesis 1.** *Healthcare expenditure and the number of medical doctors as social factors are positively correlated to population life expectancy*.

**Hypothesis 2.** *Economic development measured by GDP per capita is directly (positively) associated with life expectancy*.

**Hypothesis 3.** *CO_2_ emissions as an environmental factor have a negative impact on life expectancy*.

## 3. Research Methodology and Data

The purpose of the analysis is to quantify the possible relationships that could exist among the following four variables in the evolution of LEB: healthcare expenditure and number of medical doctors, as social factors; the gross domestic product per capita, a macro-economic variable; and CO_2_ emissions, an environmental variable.

The methodological approach used in the analysis was based on descriptive statistics, the unit root test to examine the time series characteristics of variables, the cointegration test to observe the long-term correlations between the variables, and the estimation of regression to assess the association between the dependent variable and independent predictors, while considering both cross-sectional and time series aspects.

In this study, annual data for 13 Eastern European countries over a period of 21 years (2000–2020) were used. The variables analyzed were LEB, health expenditure (% of GDP), the number of medical doctors per 10,000 inhabitants, GDP per capita (PPP constant 2017 international $), and CO_2_ emissions (metric tons per capita).

### 3.1. Descriptive Statistics

Table 1 shows the descriptive statistics of the variables considered in our model. The average LEB for all 13 countries included in the analysis over the period of 2000–2020 is around 75 years, with a very small standard deviation of only 2.67 years (3.5%), indicating certain uniformity in the average number of years that an individual is expected to live in Eastern European countries. LEB has a range between a minimum value of 70.26 years for Estonia in 2001 and a maximum value of 81.53 years for Slovenia in 2019. A higher level of variability is found in health expenditure as a percentage of GDP with a mean of 8.26% and a standard deviation of 1.77%. The minimum percentage of GDP used for health expenditure was 4.21% in the case of the Slovak Republic (in the year 2000), and the highest percentage of GDP used for health was 12.82% in the case of the Czech Republic (in the year 2020).

The number of medical doctors per 10,000 inhabitants also has a wide range of variation from around 18 for Cyprus in 2006 to almost 45 in the case of Lithuania in 2019. For Eastern Europe in the period of 2000–2020, the average number of doctors per 10,000 inhabitants was around 31, with a large standard deviation of 6 (20%).

The second-highest level of dispersion is found in the case of GDP per capita. The average annual value in this case is 25,493 $ (PPP constant 2017 international dollars), with a standard deviation of 7575 $ (29.7%). The spread of our GDP data is from the smallest value of 8900 $ for Serbia (year 2000) to a maximum value of 41,739 $ for Cyprus in 2019.

The last explanatory variable, CO_2_ emissions (metric tons per capita), has an average value of 6.50 tons per person. According to the literature insights, there is a strong relationship between GDP per capita as an indicator of economic growth and environmental pollution expressed by per capita CO_2_ emissions. However, reality shows that there can be large differences in per capita emissions, even between countries with similar living standards. For our sample of 13 Eastern European, countries we have the highest level of variation in CO_2_ emissions starting from 2.93 metric tons per capita for Latvia in 2000 to a maximum of 14.74 for Estonia in 2007. The standard deviation of CO_2_ emissions is 2.58 metric tons per capita (39.7%), the highest compared to the other three exogenous variables presented earlier in the article.

The analysis of the average values of the five variables considered for each year within the period of 2000–2020 (Figure 1) shows a slight upward trend for the dependent variable, LEB, starting from an average value of 72.7 years in 2020, following a linear increase to a value of 77.6 years in 2019 and then a slight reduction to 76.8 years in 2020. This trend was accompanied by similar evolutions of the first three predictors (health expenditure, the number of medical doctors per 10,000 inhabitants, and GDP per capita) and a slow downward evolution for the fourth independent variable (CO_2_ emissions).

All 13 Eastern European countries experienced gains in life expectancy during the period of 2000–2019 but, starting with 2019, we noticed a reduction, with possible roots in the outbreak of the COVID-19 pandemic. This phenomenon of the decrease in average life expectancy due to the constant increase in mortality during the COVID-19 pandemic has been the subject of a significant number of scientific studies. For example, in Eastern Europe, Bulgaria recorded the biggest decline in life expectancy (17.8 months in 2020 compared to 2019), thus becoming the most severe example of the evolution of this phenomenon [36].

By determining the average growth rates of the indicators in the case of each country for the entire period of 2000–2020 (Table 2), we observed increases in the case of the first four indicators and mainly decreases in the case of the fifth (CO_2_ emissions).

The biggest increases in average life expectancy in 2020 compared to 2000 were recorded by Estonia (11.6%), Latvia (6.9%), and Slovenia and Croatia, each with 6.8%. In the case of Estonia, the positive evolution of the average life expectancy is correlated with an important increase in health spending (57.6%), GDP per capita (102.0%), number of doctors per 10,000 inhabitants (31.3%), and a substantial reduction in CO_2_ emissions per capita (−50.1%). Furthermore, in Croatia, the increase in average life expectancy by 6.8% was accompanied by a 52.5% increase in health expenditure and a 54.6% increase in the number of doctors per 10,000 inhabitants, GDP per capita (47.0%) and with a small reduction in CO_2_ emissions per capita (−4.3%).

The lowest increases in life expectancy were recorded by Bulgaria (2.8%), the country that was strongly negatively affected by the COVID-19 pandemic, Poland (3.7%), Serbia (4.0%), and Lithuania (4.1%). In the case of Serbia, the very low increase in average life expectancy (4.0%) was associated with a below-average increase in health spending (22.7%) and the number of doctors per 10,000 inhabitants (20.3%) and an increase in CO_2_ emissions per capita (14.7%). Lithuania, also with a very low increase in average life expectancy (4.1%) recorded small increases in health expenditure and the number of doctors per 10,000 inhabitants (27.3%, respectively 26.0%) and the highest increase in the level of CO_2_ emissions per capita in Eastern Europe (39.2%).

The descriptive analysis of the data outlines a series of possible relationships between the average LEB and the four predictors, representing a first step in the analysis of the proposed hypotheses.

The study is based on the collection of data for five variables across the time span from 2000 to 2020, with 21 yearly observations for each variable. Our research employed empirical analysis in order to ensure consistency, including several techniques such as unit root testing and regression analysis. After investigating the specialized literature, panel data regression analysis was determined to best fit this study due to the investigation of a number of different countries with data for each measurement.

According to Baltagi and Hsiao, panel data analyses are characterized by a number of advantages and weaknesses [37,38]. The main benefits were as follows: (a) the panel analysis allows more efficient processing of information by mixing time series and cross-sectional data, leading to an increase in statistical power; (b) the application of panel data models with fixed or random effects leads to the improvement of the estimation by controlling for the observed heterogeneity; (c) panel data enables the easy examination of dynamics and changes over time by recording individual trajectories and trends; (d) and panel data analysis can improve the identification of causal relationships by controlling for unobserved variables and individual effects.

Among the drawbacks of panel data models, it is important to note the difficulty in gathering a steady amount of accurate and trustworthy data over extended periods of time, as well as the high cost involved. Selective participation can lead to selection bias, which can negatively affect representativeness in panel data models and thus bias the estimated results.

### 3.2. Panel Unit Root Test

When analyzing the individual and temporal features of the time series characteristics of variables in a panel dataset, panel unit root tests are an essential tool. These tests aim to determine whether the variables under analysis have a unit root, a sign of non-stationary. A frequently used panel unit root test is the Levin-Lin-Chu (LLC) test, which is an expansion of the Dickey and Fuller test for panel data. The LLC test takes into account individual-specific intercepts and trends; therefore, it may also accommodate cross-sectional variability using the Fisher tests and the Im-Pesaran and Shin test—IPS [39]. The existence of a unit root, or non-stationary series, is the null hypothesis of the ADF test, and stationarity is the alternative hypothesis.

For every variable, a panel regression model, including a temporal trend, lagged levels, and lagged differences, is built in order to run the LLC test. The null hypothesis, which asserts the existence of a unit root in the series and consequently non-stationary, is then compared to critical values obtained from the LLC distribution to see whether it may be rejected in favor of the alternative hypothesis of stationarity. Table 3 displays the results of the unit root tests for our variables.

Table 3 shows that since the value of *p* for each variable is smaller than the level of significance, we may reject the null hypothesis that a unit root exists in the series at the first difference and conclude that the variables are stationary.

By observing the relationships between the variables, the panel cointegration test was run in order to support the empirical analysis. A fundamental analytical technique for investigating cointegration patterns in a panel dataset (Table 4), the Pedroni panel cointegration test is widely used [40]. For our analysis, we employed Pedroni residual panel cointegration, which is based on seven panel cointegration test statistics, four based on within-dimension factors and three on between-dimension factors. Cross-sectional dependency and heterogeneity, which are crucial factors to take into account when working with panel datasets, are incorporated into the suggested methodology, which builds upon traditional cointegration analysis.

### 3.3. Estimation of the Regression Model

The Pooled Ordinary Least Square (POLS) approach is used as the initial stage in estimating a panel regression model. This method assumes that all data in a cross-section (country) is homogeneous and does not treat individual cross-sections differently [41]. With pooled OLS, this analysis does not differentiate between the 13 countries and ignores both the cross-sectional and time-series nature of the data.

The general form of OLS regression equation for pooled data [42] is as follows:$$Y_{it}=X_{it}\;\cdot \;\beta +\alpha +\varepsilon_{it}$$
in which: *i* = 1, 2, …, 13 and *t* = 1, 2, …, 21 are the number of cross-sectional units (13 *countries*) and the number of time periods (21 *years*) and $\varepsilon_{it}$ is the error term. In a Fixed Effect Model (FEM), the influence of the factor variable under consideration ($X_{it}$) on the dependent variable ($Y_{it}$) is assumed to be the same for all cross-sectional units within the analysis period.

The fixed-effects regression model equation is presented as follows:$$Y_{it}=X_{it}\;\cdot \;\beta +\alpha_{i}+\varepsilon_{it}$$
with *t* = 1, 2, …, *T* time periods (*years*) and *i* = 1, 2, …, *N* cross-sectional units (*countries*).

In order to estimate the parameters of the FEM we may consider the individual and time specificity by introducing specific effects (fixed effects) in cross-sections and periods that represent the coefficients that will be estimated.

$\alpha_{i}$ contains omitted variables that are constant over time for each unit *i*. These are called fixed effects and introduce unobserved heterogeneity into the model. $X_{it}$ is the observed part of the heterogeneity and $\varepsilon_{it}$ contains the remaining omitted variables.

We used the Hausman test (H_0_ indicates that the preferred model is random effects, whereas the alternative model is fixed effects) to evaluate the cross-sectional random effects and to identify the optimal technique between fixed effects and random effects estimators. To test the cross-sectional random effects and to determine the best method between fixed-effects and random-effects estimators, we applied the Hausman test (*H*_0_ states that the preferred model is random effects, whereas the alternative model is fixed effects). The null hypothesis is that the individual effects are not correlated with the *X*’*s* (*H*_0_: both estimates are consistent, but random effects estimates are efficient, and H_1_: fixed-effects estimates are consistent, but random effects estimates are not).

The robustness of the model should be tested using another estimation version as a random time effect hypothesis (Hausman test). Thus, we will test for correlated random effects, the assumption being that the random effect $\alpha_{i}$ needs to be uncorrelated with the other regressors (*X* variables). If not, there is an endogeneity problem, and the random effect estimator is inconsistent. The Hausman test compares two estimators: the fixed effect, which is always consistent, and the random effect estimator (consistent under $H_{0}$). Therefore, for $H_{0}$, both estimates are consistent, but the random effects estimate is efficient, and for $H_{1}$, the fixed-effects estimate is consistent, but the random effects estimate is not. If we do not have enough data to reject H_0_, the random effects model will be preferred. If we are able to reject the null hypothesis, we will choose instead the fixed-effects model.

The probability obtained for the Hausman test was smaller than the significance level, as indicated in Table 5, which led to the conclusion that the FEM would be the most suitable regression model for our study.

As a result, in our investigation, we chose a panel regression model with fixed effects.

All the independent variables, health expenditure, number of medical doctors per 10,000 inhabitants, GDP per capita, and CO_2_ emissions per capita, significantly help explain the average number of years a newborn would live as LEB. The variables are highly significant—as their *p*-values are below 0.01—and positive for the first three variables, highlighting a positive correlation with life expectancy, except for the case of CO_2_ emissions, suggesting that higher carbon emissions are capable of reducing longevity and have a negative impact on life expectancy.

As can be seen from Table 6, a one percent increase in health expenditure, as a percentage of GDP, increases LEB by 0.44 years, whereas a one-unit increase in the number of medical doctors per 10,000 inhabitants increases LEB by 0.0984 years. At the same time, each additional 10,000 USD per capita each year would increase LEB by 2 years on average. If CO_2_ emissions increase by 1 metric ton per capita, LEB would decrease by 0.2607 years.

Given that in the last 30 years we have experienced an ever-increasing economic and social integration of EU countries, which determines major interdependencies between cross-sectional units, we will test our fixed-effects model for the presence of cross-sectional dependence. Because the time dimension (T) of our panel is larger than the cross-sectional dimension (N), to test the presence of cross-sectional dependence, we will use the LM test, developed by Breusch and Pagan [43], with the null hypothesis of no cross-sectional dependence (correlation) in weighted residuals.

As can be seen from Table 7, both tests reject the null hypothesis of no cross-sectional dependence (*p* = 0.0000). Thus, we can presume that in our fixed-effects model, the cross-sectional dependence between the series is present.

### 3.4. Re-Estimated Model with GLS and SUR Weights

In econometrics, the traditional model of error cross-sectional dependency is the Seemingly Unrelated Regressions (SUR) technique, developed by Zellner [44].

When the same regressors are added to the model for each individual, the model is in the following form:$$Y_{it}=X_{it}\;\cdot \;\beta_{i}+\eta_{i}+\upsilon_{it}$$
where $\beta_{i}$ and $\eta_{i}$ are considered fixed. This approach is based on two assumptions: first, that is, all regressors in the specified model continue to be highly exogenous, and second, the asymptotics are fixed *N* and T→∞.

By using OLS at the initial stage of each individual-specific equation, the SUR technique yields a workable GLS estimator. This estimator includes consistent estimates of the parameters, such as the $N\;\cdot \;\left(N+1\right)=2$ separate entries in the error covariance matrix. As a result, the resultant $\beta_{i}$ estimator is asymptotically efficient and consistent.

A particular degree of correlation between the regression model’s residuals was taken into consideration when estimating the linear regression model’s unknown parameters using the EGLS approach. The GLS technique entails estimating a single equation that combines time series and cross-sectional data into a single column, incorporating all variables. Since autocorrelation or heteroscedasticity contradicts the fundamental tenet of ordinary least squares (OLS), which holds that the error terms are uncorrelated and have constant variance, generalized least squares (GLS) was considered a potential solution. The goal of the pooled GLS estimator is to reduce the overall sum of squared errors while taking serial correlation and heteroscedasticity into consideration. Regression analysis, which takes into account both cross-sectional and time series features, is used to evaluate the relationship between a dependent variable and independent factors. The benefits of data pooling are integrated into this methodology, which also addresses the problems of heteroscedasticity and possible serial correlation.

SUR (seemingly unrelated regression) suggests a method for estimating panel data models with a large number of time periods but few cross-sectional units. GLS improves statistical efficiency and reduces the risk of drawing inadequate inferences compared to weighted least squares methods and conventional least squares [45].

The equation for our model, re-estimated with GLS and SUR weights, takes the following form:$$\begin{array}{cc} LIFE\;EXPECTANCY & =0.37579*HEALTH\;EXPENDITURE+0.08775*NO\;DOCTORS+0.00018\\ & *GDP\;PER\;CAPITA-0.23972*CO_{2}\;EMISSIONS+66.59989+\left[CX=F\right] \end{array}$$

Table 8 displays the equation results.

Results of the final model show that all coefficients are significant at the 0.01 level. As expected, health expenditure, the number of medical doctors per 10,000 inhabitants, and GDP per capita have a positive impact on LEB, while CO_2_ emissions negatively influence the general state of health of the population and, implicitly, the average level of its longevity. We will test our fixed effects using GLS with SUR-weights model for the presence of cross-sectional dependence using Breusch–Pagan LM and Pesaran CD tests.

As can be seen from Table 9, both tests fail to reject the null hypothesis of no cross-sectional dependence (*p* = 1.00 and *p* = 0.76). Thus, we can presume that in our fixed-effects model with LEB as the dependent variable using GLS with SUR weights, the cross-sectional dependence is no longer present.

The correlation coefficients between the independent variables in our model are in the range [−0.4, +0.4], and the VIFs are between the values 1.09 and 1.32, aspects that show that multicollinearity is not a problem in our case.

## 4. Empirical Results

In order to obtain a more comprehensive view of the data utilized in the analysis, the descriptive statistics for the variables employed were first applied. The results of the Jarque–Bera test for normality showed that, for four of the five variables—LEB, health spending, GDP per capita, and CO_2_ per capita—the null hypothesis of normality was rejected. To verify the stationarity of the series and unit roots, the unit root test for the panel of data is the first step in our approach.

Hypothesis testing was used to investigate the existence of a unit root both at the series level and after the first difference. Confidence intervals of 1%, 5%, and 10% were used in this analysis. All variables show stationarity after taking the first difference, suggesting that the data series is integrated at order one. The results of the unit root test demonstrate that the time series is stationary and that the data do not have a unit root. Over the course of the whole time period, the mean and variation of stationary time series remain constant. If we take into account a 1% significance threshold, the cointegration test results show that the null hypothesis, which assumes no cointegration, is rejected when taking into account the deterministic intercept and trend for five tests. We note that the degree of testing is high, and the study supported the cointegration of the variables.

The empirical analysis was carried out using the pool estimation of the regression equation based on the findings of the unit root test, which showed that the time series is stationary. A stationary time series shows a constant mean and variation throughout the course of the entire period. The selected time series lacks seasonal characteristics and long-term trends, making it suitable for modeling and analysis.

The panel cointegration test findings demonstrate the presence of cointegration and a strong relationship between the variables in the panel dataset.

The least squares method, which minimizes the total squared differences between the model’s predicted values and the actual values, is a widely used statistical analysis methodology for estimating the parameters of a linear model. In order to address the cross-sectional dependence that existed in our first FE approach, we built the GLS with the SUR weights model using Zellner’s Seemingly Unrelated Regression (SUR) technique.

## 5. Discussion

LEB is recognized as a very important parameter of social and economic development, summarizing in the most elegant way the level of well-being in a country [7,31,46]. LEB is positively influenced by some socioeconomic factors, like economic development, higher living standards, better education, as well as greater access to quality health services. LEB is the average number of years that an individual is expected to live from birth, assuming constant age-specific mortality levels [27,47]. It is one of the most relevant determinants of a country’s well-being, which can have a major impact on other social and economic aspects such as female fertility, investment in human capital, consumer behavior, expenses regarding the pension system, and last but not least, economic growth [6]. According to specialized literature, the main factors influencing life expectancy can be summarized in three major categories: economic, social, and environmental [29].

In the 1960s, an academic debate began on the relationship between growth and social development. The debate about economic and social development particularly focused on which of these two processes precedes the other, whether economic growth is a cause of social development or whether social development precedes economic growth. Consequently, literature has been divided, over the years, into four currents of thought on the causal relationship between economic growth and social development [48,49].

This study is one of the few studies which have investigated the effects of GDP per capita (as an economic factor), health care expenditure and the number of medical doctors (as social factors), and CO_2_ emissions (as an environmental factor) on life expectancy using panel data analysis for Eastern European countries over the 2000–2020 period. Our results are stable and largely confirm the working hypotheses.

Our article analyzes the impact of several potential predictors on LEB in 13 Eastern European countries over the period of 2000–2020. We formulated three research hypotheses associated with the variables that we identified from the economic literature. Three types of models were evaluated in our paper (the POLS model, the fixed-effects model, and the random effects model). After applying several specific statistical evaluation tests (Chow and Hausman), we concluded that using the cross-country fixed-effects model could be appropriate for our data.

This article argues in favor of using the fixed time effects model in the case of LEB as the dependent variable and considering its potential predictors as independent variables such as health care expenditure and number of medical doctors (as social factors), GDP per capita (as an economic factor), and CO_2_ emissions (as an environmental factor).

As previous studies have shown [28,50], this paper confirms the relevance of healthcare expenditure and the number of medical doctors, as a proxy for healthcare system, in positively influencing a country’s population life expectancy. Our results show also that economic development, measured by GDP per capita, has a significant positive impact on LEB. It is already well known that inflation and unemployment are two major economic factors that negatively affect the LEB [51].

Our results also showed that CO_2_ emissions had a significantly negative correlation with LEB, suggesting that higher carbon emissions are capable of reducing longevity, being responsible for the increased human mortality in the last decades, as previously suggested by other studies [50].

Considering the coefficients estimated by the final fixed effects using GLS with SUR-weights model, we conclude that a one percent increase in health expenditure (as % of GDP) increases LEB by 0.376 years, whereas each additional medical doctor per 10,000 inhabitants increases LEB by 0.088 years on average. At the same time, each additional 10,000 USD per capita each year would increase LEB by 1.8 years on average. On the other hand, if CO_2_ emissions increase by 1 metric ton per capita, LEB would decrease by 0.24 years, suggesting that higher carbon emissions are capable of reducing longevity.

The *R^2^* coefficient is 0.99 (adjusted *R^2^* also 0.99), which expresses that 99% of LEB could be explained by the following exogenous factors: healthcare expenditure, the number of medical doctors, GDP per capita, and CO_2_ emissions per capita.

Our working hypotheses were completely validated. The proposed model adequately represented the main influencing factors, both with positive and negative impact, on the average LEB in Eastern Europe for the period of 2000–2020. The results of our panel regression model for Eastern European countries were also well integrated into the specialized literature regarding the determinants of LEB.

This research has a few limitations. First, although we collected the data from the databases of reliable organizations (e.g., the World Bank and World Health Organization), the presence of errors cannot be excluded. Second, the variable selection stage was influenced by data availability during the study period. Therefore, some variables that were potentially relevant to our study could not be included in our research.

Our study describes the situation of an average Eastern European country. In the near future, we intend to diversify our research by considering all European countries, and we also consider investigating this issue further by considering several subsamples separately (Western and Eastern European countries). Since we observed a strong decrease in life expectancy during the COVID-19 pandemic period, our future research will also analyze more recently available data regarding this aspect.

## 6. Conclusions

Our contribution to the existing literature is the development of an econometric model for the Eastern European region. We introduce a set of variables related to health, economic, and environmental aspects that explain most of the life expectancy trends in recent years. European countries must make special efforts to increase the life expectancy of their populations through economic and health policies aimed at the well-being of their citizens. Our results may be biased by possible errors in the analyzed databases and by the availability of the data during the study period.

This article can represent an alarm signal on the importance of reforming the medical system and also on the need to develop a legislative process related to improving the quality of the environment in Eastern European countries. Furthermore, our paper outlines the idea that the growth of GDP per capita, as a measure of socioeconomic development, plays an essential role in improving the standard of living, contributing to a longer and healthier life.

## Figures and Tables

**Figure 1 healthcare-12-01148-f001:**
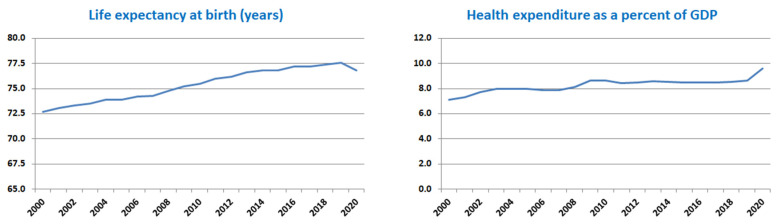
Average values of the variables for each year (all countries).

**Table 1 healthcare-12-01148-t001:** Descriptive statistics.

	Variable	Life Expectancy (Years)	Health Expenditure (% of GDP)	Doctors per 10,000 Inhabitants	GDP per Capita (PPP Constant 2017 International $)	CO_2_ Emissions (Metric Tons per Capita)
Indicator						
Mean		75.37637	8.264568	30.43051	25,493.04	6.501881
Median		75.17317	8.525605	30.97	25,423.49	6.124991
Maximum		81.52927	12.82249	44.81	41,739.46	14.74312
Minimum		70.25854	4.208762	18.2	8900.611	2.926895
Std. Dev.		2.66885	1.76676	6.087304	7574.81	2.583164
Skewness		0.309671	−0.22039	0.069325	0.004	1.049718
Kurtosis		2.510469	2.310158	2.397831	2.210982	3.635738
Jarque–Bera		7.089168	7.623068	4.34334	7.082235	54.7342
Probability		0.028881	0.022114	0.113987	0.028981	0.0000
Observations		273	273	273	273	273
Cross-sections		13	13	13	13	13

Source: Authors’ calculations using EViews.

**Table 2 healthcare-12-01148-t002:** Average growth rates of the variables for each country in the period of 2000–2020.

Country	Overall Change 2020 vs. 2000				
	Life Expectancy	Health Expenditure (% of GDP)	Doctors per 10,000 Inhabitants	GDP per Capita	CO_2_ Emissions per Capita
Bulgaria	2.8%	29.4%	26.4%	114.0%	−7.4%
Croatia	6.8%	52.5%	54.6%	47.0%	−4.3%
Cyprus	6.3%	62.1%	69.9%	19.8%	−27.2%
Czech Republic	4.3%	29.7%	23.0%	54.7%	−31.7%
Estonia	11.6%	57.6%	31.3%	102.0%	−50.1%
Hungary	6.1%	31.3%	16.9%	60.9%	−14.2%
Latvia	6.9%	7.4%	17.4%	135.3%	24.6%
Lithuania	4.1%	27.3%	26.0%	168.4%	39.2%
Poland	3.7%	44.5%	68.2%	101.2%	−4.7%
Romania	4.3%	22.8%	45.2%	138.6%	−10.1%
Serbia	4.0%	22.7%	20.3%	105.2%	14.7%
Slovak Republic	5.2%	49.0%	13.3%	97.6%	−24.7%
Slovenia	6.8%	34.3%	52.1%	41.2%	−18.8%

Source: Authors’ calculations.

**Table 3 healthcare-12-01148-t003:** Panel unit root tests.

	Tests	LLC * t Stat	IPS ** W Stat	ADF ** Fisher χ^2^	PP ** Fisher χ^2^
Variables					
Life expectancy	Level	0.000	0.199	0.336	0.177
	1st Diff.	0.000	0.029	0.000	0.000
Health expenditure	Level	0.500	0.890	0.594	0.666
	1st Diff.	0.000	0.000	0.000	0.000
Doctors per 1000 inhabitants	Level	0.971	1.000	0.778	0.584
	1st Diff.	0.000	0.000	0.000	0.000
GDP per capita	Level	0.007	0.649	0.686	0.882
	1st Diff.	0.000	0.000	0.000	0.000
CO_2_ per capita	Level	0.801	0.765	0.401	0.441
	1st Diff.	0.000	0.000	0.000	0.000

Null hypothesis H_0_: Unit root (* assumes common unit root process/** assumes individual unit root process). Source: Authors’ calculations using EViews.

**Table 4 healthcare-12-01148-t004:** Panel cointegration test.

Dimension	Statistic Test	Statistic	Probability
Within dimension (common AR coefficients)	Panel v-statistic	9.4093	0.0000
	Panel rho-statistic	0.0553	0.5220
	Panel PP-statistic	−3.6436	0.0001
	Panel ADF-statistic	−3.8209	0.0001
Between dimension (individual AR coefficients)	Group rho-statistic	1.4804	0.9306
	Group PP-statistic	−5.9085	0.0000
	Group ADF-statistic	−5.2431	0.0000

Null hypothesis H_0_: No cointegration, trend assumption: Deterministic intercept and trend. Source: Authors’ calculations using EViews.

**Table 5 healthcare-12-01148-t005:** Hausman test (correlated random effects).

Test Summary	Chi-Square Statistic	d.f.	Probability
Cross-section random	22.271	4	0.000

Source: Authors’ calculations using EViews.

**Table 6 healthcare-12-01148-t006:** A fixed-effects model with LEB as the dependent variable. Dependent variable: LEB.

Variable	Coefficient	Std. Error	t-Statistic	Probability
Health expenditure	0.43795	0.09725	4.50335	0.0000
Doctors per 1000 inhabitants	0.09840	0.02897	3.39694	0.0008
GDP per capita	0.00021	0.00002	12.07829	0.0000
CO_2_ per capita	−0.26070	0.06844	−3.80936	0.0002
R-squared	0.9055			
Adjusted R-squared	0.8996			
F-statistic	153.3888			
Probability (F-statistic)	0.0000			

Source: Authors’ calculations using EViews.

**Table 7 healthcare-12-01148-t007:** Residual cross-section dependence tests.

Test	Statistic	d.f.	Prob.
Breusch–Pagan LM	500.81260	78	0.0000
Pesaran CD	18.90647	-	0.0000

Null hypothesis: No cross-section dependence (correlation) in residuals. Source: Authors’ calculations using EViews.

**Table 8 healthcare-12-01148-t008:** A fixed-effects model with LEB as the dependent variable using GLS with SUR weights. Dependent variable: LEB (panel EGLS cross-section SUR).

Variable	Coefficient	Std. Error	t-Statistic	Probability
Health expenditure	0.37579	0.02405	15.62495	0.0000
Doctors per 1000 inhabitants	0.08775	0.00498	17.62622	0.0000
GDP per capita	0.00018	0.00001	18.91017	0.0000
CO_2_ per capita	−0.23972	0.01641	−14.60537	0.0000
R-squared	0.9906			
Adjusted R-squared	0.9900			
F-statistic	1691.1210			
Probability (F-statistic)	0.0000			

Source: Authors’ calculations using EViews.

**Table 9 healthcare-12-01148-t009:** Residual cross-section dependence tests.

Test	Statistic	d.f.	Prob.
Breusch–Pagan LM	13.80523	78	1.0000
Pesaran CD	0.30217	-	0.7625

Null hypothesis: No cross-section dependence (correlation) in weighted residuals. Source: Authors’ calculations using EViews.

## Data Availability

Dataset available on request from the authors.

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
