# Peer review of "Social and Economic Determinants of Life Expectancy at Birth in Eastern Europe"

_healthcare, 2024, doi:10.3390/healthcare12111148_

Round 1
Reviewer 1 Report
Comments and Suggestions for Authors
Comments
1) The paper seems to study the effects of select economic factors, social factors environmental factors on life expectancy at birth in the context of select 13 Eastern European countries. The authors state that “Our study addresses this assumption by surveying 13 Eastern European countries…..”. Which assumption do the authors address? It is not communicated to prospective readers. [109-111]
2) Why do authors conduct the study herein in the context of select Eastern European countries? Is extant literature on the topic scarce or left entirely unaddressed in the Eastern European context? [General Comment]
3) The authors are advised in their way of stating research hypotheses. For instance, the second hypothesis gives no direction as to what authors expect as to the select economic factor’s impact on the dependent variable. Positive association? Neutral? Negative? It needs to be explicitly stated. [173-179]
4) Table 6 presents the results of the authors’ fixed effects estimation. I have failed to see within R-squared and between R-squared values. In the context of fixed effects estimation, these values would be informative when it comes to evaluating the goodness of fit of the model that is subject to estimation [392-398]
5) I have failed to see Variance Inflation Factor values as to the select variables so that we could understand if there are issues as to multicollinearity. For instance, just by definition, GDP per capita and health expenditures as a percentage of GDP are correlated. Multicollinearity tends to inflate R-squared values and presented values seem unusually high [121-140] & [465-468] & [394-398]
Author Response
Dear Reviewer 1,
Thank you very much for evaluating our manuscript. Your recommendations and comments have helped us improve our manuscript. Here we provide the requested corrections and address the comments. The changes we have made in the manuscript are highlighted in red.
- “Our study addresses the effects of economic factors, social factors and environmental factors on LEB” - We clarified this aspect at the end of the Introduction chapter.
- Besides the fact that the specialized literature with reference to Eastern European countries is poor in terms of life expectancy at birth, we chose for analysis the 13 countries due to their common recent history (the last 50 years), these countries being part of the former communist
- We expect a direct - positive correlation between GDP per capita and LEB. We corrected the text, thank you.
- In the initial fixed effects model, R-square was 0.9055 (Table 6), but after re-estimation of the model with GLS and SUR weights we obtained an adjusted R-square of 0.99 (Table 8).
- The correlation coefficients between the independent variables in our model are in the range [-0.4, +0.4], and the VIFs are between the values ​​1.09 and 1.32, aspects that show that multicollinearity is not a problem in our case. We introduced this paragraph in the main text of the manuscript (paragraph 3.4).
Thank you again for reviewing our manuscript,
Reviewer 2 Report
Comments and Suggestions for Authors
The manuscript healthcare-3010102 describes the results of an analysis of life expectancy at birth in Eastern Europe between 2000 and 2020 and factors from the environment, economy, and healthcare variables that can influence it. The authors demonstrate that GDP, density of physicians, healthcare expenditure, and lower carbon dioxide emissions improve life expectancy at birth with variable impact. The authors conclude that Eastern European countries need to make a specific effort at improving their healthcare system, increase expenditure and decrease overall CO2 emissions to improve their populations' life expectancy and overall well-being. GDP as a marker for socio-economic health also needs to be taken in consideration when analyzing these data.
Comments:
Study design: The study is well designed and analyzes publicly available datasets and looks at impact of variables such as GDP, density of physicians, CO2 emissions on life expectancy at birth. There is no direct cause-effect / dose-response impact, but more associations. The hypothesis and study questions are well delineated and clear. The overall manuscript may benefit from some restructuring, including materials and methods and results sections being clearly delineated.
Introduction and discussion: The introduction is very thoroughly written, however could be significantly shortened. The discussion is systematically written and some of the introduction points could be included in the discussion.
Results: the results are very thoroughly described and may be summarized more, as some of the data and methodology could be included in methods. Given the significant amount of data, some of the tables could be included in supplemental table or figures? Their models are very interesting, however the description feels very long. Table 1 should include the units of Life expectancy, Health expenditure, GDP and CO2 emissions in the title of the respective column.
Minor comments:
Several spelling errors should be corrected, and the manuscript may benefit from review and some editing by a native English-speaking person.
Comments on the Quality of English Language
Several spelling errors should be corrected, and the manuscript may benefit from review and some editing by a native English-speaking person.
Author Response
Dear Reviewer 2,
Thank you very much for evaluating our manuscript. Your recommendations and comments have helped us improve our manuscript. Here we provide the requested corrections and address the comments. The changes we have made in the manuscript are highlighted in red.
- We made some changes to the structure of the manuscript and moved some of the introductory paragraphs to the Discussions chapter.
- We corrected Table 1. Regarding the Methodology and Results, we preferred to leave all the tables and descriptions in the main text, for a better transparency of the study.
- We corrected English language.
Thank you again for reviewing our manuscript,
Reviewer 3 Report
Comments and Suggestions for Authors
The manuscript presented for review is an interesting analysis of various socio-economic and environmental factors on life expectancy at birth. The study is well designed and all methods and experiments are presented clearly. The results are well described and support the conclusions of the article.
My only concerns are as follows:
The Introduction section should contain more general information on the reasoning for choosing these particular factors influencing life expectancy at birth. As it stands it contains a lot of information that should be placed in Discussion.
I do not quite understand the reasoning behind choosing CO2 emissions as an influencing factor. All the others are socio-economic. CO2 seems a bit out of place.
Minor English proofing needed.
Good luck.
Comments on the Quality of English LanguageMinor editing needed
Author Response
Dear Reviewer 3,
Thank you very much for evaluating our manuscript. Your recommendations and comments have helped us improve our manuscript. Here we provide the requested corrections and address the comments. The changes we have made in the manuscript are highlighted in red.
- We made some changes to the structure of the manuscript and moved some of the introductory paragraphs to the Discussions chapter.
- Since LEB is a demographic characteristic with multifactorial determinism, we considered the analysis of multiple socio-economic factors, as well as the CO2 emission as an expression of the degree of industrialization of Eastern European countries during the last 20 years. We added this paragraph in the introduction, at the aim of the study section.
Thank you again for reviewing our manuscript,